# An Examination of the Anti-Cancer Properties of Plant Cannabinoids in Preclinical Models of Mesothelioma

**DOI:** 10.3390/cancers14153813

**Published:** 2022-08-05

**Authors:** Emily K. Colvin, Amanda L. Hudson, Lyndsey L. Anderson, Ramyashree Prasanna Kumar, Iain S. McGregor, Viive M. Howell, Jonathon C. Arnold

**Affiliations:** 1Bill Walsh Translational Cancer Research Laboratory, Kolling Institute, St Leonards 2065, Australia; 2School of Medical Sciences, Faculty of Medicine and Health, University of Sydney, Sydney 2006, Australia; 3Lambert Initiative for Cannabinoid Therapeutics, University of Sydney, Sydney 2050, Australia; 4Department of Pharmacology, Sydney Pharmacy School, University of Sydney, Sydney 2006, Australia; 5Brain and Mind Centre, University of Sydney, Sydney 2050, Australia; 6School of Biotechnology and Biomolecular Sciences, University of New South Wales, Sydney 2052, Australia

**Keywords:** mesothelioma, cannabinoids, cannabidiol, cannabigerol, anti-proliferative, apoptosis

## Abstract

**Simple Summary:**

Mesothelioma is a deadly disease with few treatment options. Phytocannabinoids derived from the cannabis plant are garnering interest for their anti-cancer properties, however very little is known about their effects in mesothelioma. We aimed to assess whether phytocannabinoids have anti-cancer effects in mesothelioma and potential modes of action. We showed that several phytocannabinoids inhibited growth of mesothelioma cells, with two phytocannabinoids, cannabidiol (CBD) and cannabigerol (CBG), being the most potent. CBD and CBG also inhibited mesothelioma cell migration and invasion. Gene expression analysis highlighted signalling pathways that play a role in how CBD and CBG may exert their anti-cancer effects. CBD and CBG were unable to increase survival in a rat model of mesothelioma but this may be due to limitations in the drug delivery method.

**Abstract:**

Mesothelioma is an aggressive cancer with limited treatment options and a poor prognosis. Phytocannabinoids possess anti-tumour and palliative properties in multiple cancers, however their effects in mesothelioma are unknown. We investigated the anti-cancer effects and potential mechanisms of action for several phytocannabinoids in mesothelioma cell lines. A panel of 13 phytocannabinoids inhibited growth of human (MSTO and H2452) and rat (II-45) mesothelioma cells in vitro, and cannabidiol (CBD) and cannabigerol (CBG) were the most potent compounds. Treatment with CBD or CBG resulted in G0/G1 arrest, delayed entry into S phase and induced apoptosis. CBD and CBG also significantly reduced mesothelioma cell migration and invasion. These effects were supported by changes in the expression of genes associated with the cell cycle, proliferation, and cell movement following CBD or CBG treatment. Gene expression levels of *CNR1*, *GPR55*, and *5HT1A* also increased with CBD or CBG treatment. However, treatment with CBD or CBG in a syngeneic orthotopic rat mesothelioma model was unable to increase survival. Our data show that cannabinoids have anti-cancer effects on mesothelioma cells in vitro and alternatives of drug delivery may be needed to enhance their effects in vivo.

## 1. Introduction

Malignant mesothelioma is an aggressive cancer with a poor response to current therapies and, consequently, has a very grim prognosis. The median survival for malignant mesothelioma is only approximately 12 months and the 5-year survival rate is less than 10% [1,2]. Worldwide, over 38,000 people die each year from mesothelioma [3] with asbestos exposure linked to approximately 80% of all cases [4]. Fortunately, asbestos use has now been banned in many countries but its widespread use in thousands of construction-related products for over a century has left a deadly legacy in the community. Approximately 125 million people have experienced occupational exposure to asbestos, but a non-occupational wave is also emerging [5]. This, together with the 20–40-year latency period for disease development means that mesothelioma is not a disease of the past, but rather one that is and has been increasing in incidence. As most mesothelioma patients are diagnosed when their disease is advanced, systemic chemotherapy of pemetrexed with cisplatin is the standard of care and most effective treatment option [6,7]. However, failure of this treatment is inevitable and there are no proven second line options [2]. Therefore, there is an urgent unmet need for more effective treatments.

With the legalisation of medicinal cannabis around the world there has been a renewed interest in examining the therapeutic potential of the cannabinoids. Cannabis contains over 100 plant-derived cannabinoids (phytocannabinoids), including the main psychoactive constituent ∆^9^-tetrahydrocannabinol (THC) and the non-intoxicating compound cannabidiol (CBD). Although CBD and THC are the most studied, there is growing interest in the therapeutic potential of other phytocannabinoids, for example cannabigerol (CBG) and cannabidiolic acid (CBDA) [8,9,10,11,12]. The majority of research has focused on the use of THC in the symptomatic management of cancer due to its appetite stimulatory, analgesic, and anti-emetic effects [13,14]. However, evidence suggests that the major cannabinoids CBD and THC display anti-tumoural properties in pre-clinical models and while clinical trials are still pending, anecdotal evidence for efficacy does exist [15,16]. Cannabinoids were shown to decrease the growth and metastatic potential of cancer cells in a number of solid cancers, as well as haematological malignancies (reviewed in [17,18,19]). However, no prior studies have examined the anti-cancer effects of cannabinoids against mesothelioma. The overall aim of this project is to determine whether a panel of phytocannabinoids have anti-cancer properties in pre-clinical mesothelioma models in vitro and in vivo.

## 2. Materials and Methods

Cell culture: the human mesothelioma cell lines MSTO-211H (MSTO) and NCI-H2452 (H2452) were purchased from the ATCC. MSTO and H2452 cells are both derived from pleural mesothelioma patients [20,21]. MSTO cells display biphasic sarcomatoid and epithelioid features, while H2452 cells are epithelioid. The rat mesothelioma cell line II-45 (also biphasic) was kindly provided by A/Prof. Emanuela Felley-Bosco, Zurich University [22]. All cell lines were cultured in RPMI-1640 with L-glutamine (Gibco, Thermofisher Scientific, North Ryde, NSW, Australia) supplemented with 10% foetal bovine serum at 37 °C with 5% CO_2_, unless stated otherwise. All experiments were performed at least 3 times.

Phytocannabinoids: THC, CBD, CBDA, cannabigerolic acid (CBGA), cannabidivarin (CBDV), CBG, Δ^9^-tetrahydrocannabivarin (THCV), and cannabinol (CBN) were purchased from THC Pharm GmbH (Frankfurt, Germany). Cannabichromene (CBC) was synthesised by Lambert Initiative chemists [23]. Δ^9^-tetrahydrocannabinolic acid (THCA) was isolated from hemp extracts [24]. Cannabidivarinic acid (CBDVA), cannabichromenic acid (CBCA) and cannabichromevarinic acid (CBCVA) were synthesised by Professor Michael Kassiou at the University of Sydney (Australia). For all in vitro experiments, phytocannabinoids were dissolved in ethanol and added to cells in culture medium to a maximum concentration of 0.5%.

Cell viability/cytotoxicity: cells were seeded in triplicate in 96-well plates (Nunc MicroWell 96-Well Microplates, ThermoFisher Scientific) at a density of 5 × 10^3^ cells/well for MSTO and H2452 cells, and 2 × 10^3^ cells/well for II-45 cells. After 4 h, cells were treated with serial 2-fold dilutions of phytocannabinoids. After 72 h of treatment, cell viability was measured by standard MTT (3-(4,5-dimethythiazol-2-yl)-2,5-diphenyltetrazolium bromide) assays [25].

Cell cycle assay: cell lines were seeded and treated for 24 h with the phytocannabinoids (at concentrations double that of the 72 h IC_50_; II-45 cells: CBD 15 µM and CBG 33.2 µM; MSTO cells: CBD 22 µM and CBG 32.9 µM; H2452 cells: CBD 20 µM and CBG 31 µM) or vehicle as control. After treatment, cells were harvested, washed with 1X PBS, fixed with 1 mL of ice-cold 70% ethanol and stored at −20 °C overnight. The fixed cells were stained and analysed using the MUSE^®^ Cell Cycle Kit and the MUSE^®^ Cell Analyser following the manufacturer’s instructions [26]. The percentage of cells in G0/G1, S and G2/M phases was assessed.

Apoptosis assay: cells were seeded and treated with the phytocannabinoid, or vehicle control for 24 h, as described for the cell cycle assay. After treatment, cells were harvested, stained, and analysed using the MUSE^®^ Annexin V and Dead Cell Kit and the MUSE^®^ Cell Analyser (Abacus dx, Cannon Hill, QLD, Australia) following the manufacturer’s instructions.

Transwell migration and invasion assays: the effect of phytocannabinoids on mesothelioma cell migration and invasion was assessed using transwell plates containing 8.0 μm pore size inserts (Sigma-Aldrich Pty. Ltd., Sydney, Australia). For migration assays cells were seeded in 350 µL serum free media in the upper chamber (1.25 × 10^4^ for rat II-45 and human H2452 cells and 5 × 10^4^ cells for human MSTO cells). For the invasion assays transwells pre-coated with Matrigel™ were used and cells were seeded in 500 µL serum free media in the Matrigel^TM^ coated chamber (3 × 10^4^ of rat II-45 cells and 2 × 10^5^ of human H2452 cells). Media containing 10% FBS was added as a chemotactic agent to the lower chamber. Sub-lethal concentrations of cannabinoids were then added (or vehicle only for control). After treating for 24 h (with the exception of II-45 migration assays, which were treated for 6 h), media and non-migrated cells were removed with a cotton swab. Membranes were fixed in 70% ethanol and migrated/invaded cells were stained with Prolong^TM^ Gold Antifade Mountant with DAPI (Thermofisher Scientific). Images of migrated/invaded cells were taken at 100× magnification with an Olympus IX70 fluorescence microscope (Olympus, Macquarie Park, NSW, Australia). The number of cells in 5 fields of view was calculated using CellProfiler [27] and graphs were plotted as percentage of migrated/invaded cells normalised to vehicle only control cells in GraphPad Prism (GraphPad Software, San Diego, CA, USA). It is noteworthy that MSTO cells did not invade, therefore invasion assays were unable to be performed.

RNA extraction and cDNA synthesis: II-45, MSTO and H2452 cells were seeded and left to adhere for 4 h. Cells were then treated with CBD, CBG, or a vehicle as described for the cell cycle assay. After the 24 h treatment, cells were counted, and viability measured using the MUSE^®^ Count & Viability Kit (Abacus dx) following the manufacturer’s instructions. At harvesting, cells were greater than 80% viable. RNA was extracted using an RNeasy Mini RNA Isolation kit (Qiagen, Clayton, VIC, Australia) following the manufacturer’s instructions and quantified using the Qubit RNA BR assay kit (Life Technologies Australia Pty. Ltd., Mulgrave, VIC, Australia). Quality was assessed from the 260/280 ratio using the NanoDrop Spectrophotometer only accepting ratios > 1.6. cDNA was synthesised using the Superscript IV VILO Master Mix (Thermofisher Scientific) as per the manufacturer’s instructions.

TaqMan gene expression assays: TaqMan gene expression assays (Thermofisher Scientific) were used to quantify changes in gene expression for receptors known to interact with phytocannabinoids [28,29]. Taqman assays and identification numbers are listed in the Appendix A. Assays were performed on the QuantStudio™ 12K Flex Real Time PCR System (Applied Biosystems, ThermoFisher Scientific) as per the manufacturer’s instructions. The results were analysed on QuantStudio 12K Flex Software v1.2.2. Fold change in gene expression for each target was determined using the 2^−ΔΔCt^ method normalised to the housekeeping gene (*TBP*). Fold change (FC) in gene expression was calculated relative to vehicle-treated control cells.

TaqMan^TM^ OpenArray^TM^ Real-Time-PCR: the TaqMan^TM^ OpenArray^TM^ Human Cancer Panel (Thermofisher Scientific) was used to evaluate the expression of 624 known cancer-related genes in vehicle control, CBD, and CBG-treated MSTO and H2452 cells. Taqman assays included on the OpenArray are listed in Appendix A. cDNA was prepared as described above prior to adding to the TaqMan^TM^ OpenArray^TM^ Real-time PCR Mastermix (Applied Biosystems) and distributed on a 384-well open array plate using the OpenArray^TM^ AccuFill™ System following the manufacturer’s instructions. The PCR was then run on the QuantStudio™ 12K Flex Real Time PCR System (Thermofisher Scientific). QuantStudio™ 12K Flex Software v1.2.2 (Thermofisher Scientific) was used to normalise gene expression results to a panel of housekeeping genes included on the TaqMan^TM^ OpenArray^TM^ Human Cancer Panel and calculate FC between treated versus control cells using the 2^−ΔΔCt^ method. Gene expression results can be found in Appendix A. A FC of >2 was used as the cut-off for determining differentially expressed genes (DEGs). Genes found to be FC > 2 and shared between the MSTO and H2452 cell lines are listed in Appendix A.

Pathway enrichment analysis: to better understand the biological relevance of the identified DEGs, pathway enrichment analysis was performed. Data were analysed using ingenuity pathway analysis [30]. Molecules from the dataset that had a FC > 2 and were associated with a canonical pathway in the Ingenuity Knowledge Base were considered for the analysis. The significance of the association between the dataset and the canonical pathway was measured in two ways: (1) a ratio of the number of molecules from the dataset that map to the pathway divided by the total number of molecules that map to the canonical pathway; and (2) a right-tailed Fisher’s Exact Test was used to calculate a *p*-value determining the probability that the association between the genes in the dataset and the canonical pathway is explained by chance alone. A comparison analysis was then used to identify shared canonical pathways between the MSTO and H2452 cells (−log (*p*-value) > 2). IPA z-scores predict activation or inhibition of the canonical pathways with z-scores of >2 considered significantly activated and <−2 considered significantly inhibited. Data presented represents the top shared canonical pathways with z-scores in the same direction in both cell lines, i.e., either activated or inhibited in both MSTO and H2452 cells with general cancer canonical pathways being removed for relevance. The full list of shared canonical pathways is available in Appendix A.

In vivo experiments: all procedures involving animals were carried out in accordance with the Australian Code of Practice for the Care and Use of Animals for Scientific Purposes. The protocol for this study was approved by the Royal North Shore Hospital Animal Care and Ethics Committee (protocol numbers RESP/17/30). Female Fischer 344 rats weighing 150–200 g were housed in groups of three at the Kearns Facility under standard conditions (12 h light/dark cycles, free access to food and water, nesting material with plastic huts, and environmental enrichment, such as soft wood sticks, straws, paper and tissue boxes).

Pharmacokinetic assay to determine the vehicle for the in vivo survival experiment: 20 mg/kg CBD was made up in either: (1) hemp seed oil; (2) ethanol/tween80/0.9% saline (at a 1:1:18 ratio) or DMSO/kolliphor/0.9% saline (at a 1:1:18 ratio) to determine which vehicle provided the highest plasma levels of exposure to CBD. Rats (*n* = 4 per group) were intraperitoneally (i.p.) injected with CBD in the different vehicles. Blood was taken at 15 min, 1 h, 2 h, and 4 h post i.p. injection into EDTA blood tubes. Plasma concentrations of CBD were measured using LC MS/MS and calculated as previously described [31,32]. Results are presented in Appendix A.

In vivo survival study: on day zero, female Fischer rats were pleurally engrafted with 100 μL of serum free media containing 1 × 10^4^ II-45 cells. Pharmacological treatments began on day 1 and continued for 4 weeks. The positive control, a standard of care chemotherapy combination of 1 mg/kg cisplatin and 6.7 mg/kg pemetrexed, was administered i.p. on days 3, 6, 11, 15, 20, and 24. 40 mg/kg CBD and 100 mg/kg CBG were administered with single daily i.p. injections in the vehicle of DMSO:kolliphor:saline at a 1:1:18 ratio. Deiana et al. [33] compared the plasma pharmacokinetics of CBD and CBG in rats following i.p. injections and found that total plasma exposures (AUC 0–∞) were approximately 2.5-fold less in CBG compared to CBD dosed animals. CBD and CBG doses were thus selected based on extrapolated data from our pharmacokinetic experiment and that of Deiana et al. [33] to result in similar plasma concentrations (Cmax = 2–4 µM CBD and 2 µM CBG), as well as the maximum concentration able to be diluted in solvent. This approximates the maximal plasma concentrations that have been achieved in intractable epilepsy patients taking very high, oral doses of CBD, thus representing maximal clinically relevant doses that can be attained in humans with registered pharmaceutical CBD products [34]. Control animals received the vehicle only, using the same treatment schedule as CBD and CBG. Rats were euthanised at ethically approved endpoints (weight loss of >20% or difficulty in breathing) and blood, tissue, and tumour samples were collected. Tumours were scored on a scale of 1 to 5 based on volume and infiltration of the tumour into the pleural cavity, as described previously [35], with 1 indicating the tumour occupied up to 10% of the pleural cavity and 5 indicating the tumour occupied 40–50% of the pleural cavity.

Statistical analysis: GraphPad Prism (Version 9.3.1) was used for graphs and statistical analysis. For cell viability/cytotoxicity assays, non-linear (curve fit) regression algorithms were used to calculate the drug dose causing 50% growth inhibition (IC50 drug dose). For cell cycle, apoptosis assays and TaqMan gene expression assays, two-way ANOVA with Holm-Sidak’s multiple comparisons was used. For transwell migration and invasion assays, one-way ANOVA with Kruskal-Wallis multiple comparisons was used. For survival analysis, *p*-values were calculated using Log-rank (Mantel-Cox) test, relative to control treated animals.

## 3. Results

### 3.1. Phytocannabinoids Inhibit Viability of Mesothelioma Cells In Vitro

A panel of 13 phytocannabinoids was screened for their ability to inhibit the viability of mesothelioma cell lines. All phytocannabinoids reduced the viability of both rat (II-45) and human (MSTO and H2452) mesothelioma cell lines with the drug concentration causing 50% inhibition (IC50) shown in Figure 1. In general, the neutral cannabinoids (i.e., THC, CBN, CBD, CBG, and CBC) were more potent than the acid forms of the molecules (THCA, CBDA, CBGA, and CBCA). Overall, CBD and CBG most potently reduced the viability of the mesothelioma cell lines.

### 3.2. CBD and CBG Induce G0/G1 Cell Cycle Arrest and Apoptosis in Mesothelioma Cell Lines

Having identified CBD and CBG as the most promising phytocannabinoids, assays were undertaken to ascertain whether these cannabinoids induced programmed cell death in the rat (II-45) and human (MSTO and H2452) mesothelioma cell lines. After 24 h of CBD or CBG treatment, cell cycle changes were identified with G0/G1 arrest and delayed entry into S Phase. Apoptosis was also induced by both CBD and CBG (Figure 2 and Figure 3).

### 3.3. CBD and CBG Inhibit Migration and Invasion of Mesothelioma Cell Lines In Vitro

We then assessed whether CBD and CBG could inhibit migration and invasion of mesothelioma cell lines. Although similar to migration assays, invasion assays examine the potential of the mesothelioma cells to invade through an extracellular matrix, which is extremely important to their metastatic ability. Of note, the human MSTO cells were unable to invade through an extracellular matrix, therefore invasion assays were not examined in this cell line. Using sub-cytotoxic concentrations, both CBD and CBG significantly reduced the migratory and invasive potential of the rat II-45 and human H2452 mesothelioma cell lines, suggesting that these compounds may also slow the metastatic spread of mesothelioma cells (Figure 4).

### 3.4. CBD and CBG Treatment Effects on Cannabinoid-Related Gene Targets in Mesothelioma Cell Lines

To begin to investigate mechanisms of action, we examined whether CBD and CBG treatment affected the gene expression of targets in mesothelioma cells implicated in the effects of cannabinoids, or the pathobiology of mesothelioma (Figure 5). *CNR2* was not expressed in any of the mesothelioma cell lines and was not included in the analyses. In all three mesothelioma cell lines, both CBD and CBG robustly upregulated mRNA for the cannabinoid CB1 receptor (*CNR1*), G protein-coupled receptor 55 (*GPR55*) and the 5-HT1a receptor (*HTR1A*) (Figure 5). Notably, CBD and CBG upregulated these targets by as high as approximately 50-fold compared to vehicle-treated cells. CBD and CBG increased mRNA expression of transient receptor potential vanilloid type 1 (*TRPV1*) in all mesothelioma cell lines, except in CBD-treated H2452 cells. CBD and CBG did not consistently affect *TRPV2* or peroxisome proliferator-activated receptor gamma (*PPARG*) expression. Interestingly, CBD and CBG increased expression of C-X-C chemokine receptor 4 (*CXCR4*) in all cell lines. CBD treatment was associated with downregulation of the endogenous agonist of *CXCR4*, C-X-C motif chemokine 12 (*CXCL12*), in all cell lines. Similarly CBG reduced *CXCL12* expression in the II-45 and H2452 cells but increased its expression in the MSTO cells. 

### 3.5. Identification of Genes and Pathways Associated with CBD and CBG Treatment

A more extensive analysis of the molecular impacts of CBD and CBG on human mesothelioma cell lines was performed using gene expression arrays. The complete list of differentially expressed genes are available in Appendix A. Figure 6a,b summarise the number of dysregulated genes (FC > 2) identified following CBD or CBG treatment. CBD treatment resulted in 181 and 255 dysregulated genes in the MSTO and H2452 cells, respectively. Of these, 94 were found to be shared in both cell lines. CBG treatment resulted in 184 and 223 dysregulated genes in the MSTO and H2452 cells, respectively. Of these, 116 were found to be shared in both cell lines. Appendix A lists shared genes with FC >2 in mRNA expression as a result of CBD and CBG treatment. Of note, CBD and CBG consistently decreased expression of key cell cycle genes, including those encoding cyclins (*CCB1, CCB2,* and *CCNE1*) and cyclin-dependent kinases (*CDK1* and *CDK2*), as well as the anti-apoptotic gene *BIRC5*, and pro-metastatic gene *ID1* and the *DTL* gene, involved in an array of cancer promoting pathways, including proliferation, migration, and invasion [36,37]. Additionally, CBD and CBG consistently increased expression of the pro-apoptotic gene *BBC3* and of *GDF-15*, a gene involved in numerous biological functions and a promising prognostic marker in numerous cancers (reviewed in [38]). Pathway enrichment analysis was then performed using the shared differentially expressed genes to gain mechanistic insight into how CBD and CBG affect the mesothelioma cell lines (Appendix A). Figure 6c,d summarise the top shared canonical pathways predicted to be activated or inhibited with CBD or CBG treatment. Both CBD and CBG inhibited many pathways involved in cell cycle regulation, supporting our in vitro results. Additionally, both cannabinoids activated pathways involved in regulating intracellular calcium levels (e.g., Gαq, phospholipase C (PLC) and TEC kinase signalling) and inflammatory/immune responses (e.g., thrombin and NF-κB signalling). The inhibition of DNA repair pathways (base excision repair and nucleotide excision repair) by CBG was also noted.

### 3.6. CBD and CBG Did Not Increase Survival in an Immunocompetent Rat Model of Mesothelioma

We then wished to observe whether the promising effects of CBD and CBG observed in vitro could be translated in vivo. We first aimed to determine which vehicle provided the optimal level of plasma exposure in rats using CBD as the indicative compound. 20 mg/kg CBD i.p. was administered in 3 different vehicles (hemp seed oil, ethanol/Tween80/saline or DMSO/kolliphor/saline). After injecting animals, blood was taken at 15 min, 1 h, 2 h, and 4 h post-injection and then assessed for plasma CBD concentrations using LC-MS/MS. Appendix A shows that the DMSO/kolliphor/saline vehicle provided marginally superior plasma CBD exposures with a more prolonged half-life compared to the ethanol/Tween80/saline vehicle (AUC = 130 versus 150 µg per min/mL; t_1/2_ = 112 versus 210 min for ethanol versus DMSO vehicles, respectively). Hemp seed oil resulted in very low plasma CBD concentrations. The DMSO/kolliphor/saline vehicle was thus used for the in vivo survival study.

A syngeneic orthotopic rat mesothelioma model [39] was used to assess whether daily treatment for 4 weeks with 40 mg/kg CBD or 100 mg/kg CBG i.p. compared to standard of care chemotherapy (1 mg/kg cisplatin + 6.7 mg/kg pemetrexed) improved survival (Figure 7). Although survival was the primary endpoint of this study, severity of pleural dissemination at endpoint was also assessed. No statistical differences were observed between the groups (data not shown). As expected, treatment with cisplatin + pemetrexed significantly prolonged survival compared to vehicle-treated animals (median survival 28 days vs. 23 days, respectively; *p* < 0.05), however neither CBD nor CBG prolonged survival.

## 4. Discussion

As failure of standard of care treatment is inevitable in mesothelioma, there is an urgent unmet need for more effective treatments. With the legalisation of medicinal cannabis around the world there has been an increasing focus on the plant cannabinoids and their medicinal potential, including in the treatment of cancer. Cannabinoids display anti-cancer properties, including anti-proliferative, anti-migratory, anti-invasive, and anti-angiogenic effects in preclinical cancer models (reviewed in [17,18,19]). These anti-cancer effects have been observed in an array of different cancers, including breast, prostate, brain, lymph, blood, pancreas, bladder, and skin (reviewed in [40]). However, the present study is the first to investigate anticancer effects of phytocannabinoids in preclinical mesothelioma models. We screened a panel of 13 phytocannabinoids of which CBD and CBG had the most potent and consistent anti-proliferative effects across rat and human mesothelioma cell lines. The anti-proliferative effects of CBD and CBG were associated with cell cycle arrest and apoptosis. Moreover, CBD and CBG reduced the migration and invasiveness of the mesothelioma cells in vitro. However, systemic administration of CBD and CBG in vivo at levels of exposure relevant to human dosing [31,32,34] was ineffective in prolonging the lifespan of rats in a syngeneic orthotopic mesothelioma model.

In this study, we screened a comprehensive panel of cannabinoids that are found in cannabis, some of which are abundant in specific cannabis strains. This included both acidic and neutral forms of the phytocannabinoids. The acidic forms (e.g., CBDA) are enzymatically biosynthesised in the plant and then decarboxylated via exposure to heat or air into the neutral cannabinoids (e.g., CBD). Most prior research has focused on the neutral forms, such as THC and CBD, however there is emerging evidence that the acidic compounds also have therapeutic potential [24]. Here, we show that the acidic cannabinoids display consistently lower potency than the neutral cannabinoids in reducing the proliferation of mesothelioma cells in vitro. Compounds with a carboxylic acid moiety are notorious for having poor membrane permeability, thus it is possible that the reduced anti-proliferative potency of the acids versus the neutral cannabinoids may be due to diminished access of the acid forms to intracellular anti-cancer targets [41]. This might be further exacerbated by active transport by drug efflux proteins expressed in mesothelioma cells [35,42,43], as some acidic cannabinoids appear to be substrates of ABC transporters. For example, CBDA is an excellent substrate of ABCG2 (breast cancer resistance protein), and CBCA is a substrate of ABCB1 (P-glycoprotein) [44,45].

CBD and CBG were found to be the most potent at decreasing cell viability with IC50 values within the low µm range in all three cell lines. Both these compounds do not appear to have THC-like intoxicating effects which makes them more favourable candidates for therapeutic development [11]. Hence, we focused on CBD and CBG to more comprehensively explore their anti-cancer potential and modes of action. Both phytocannabinoids showed induction of G0/G1 arrest, as well as delayed entry into S-phase, coinciding with decreased expression of key cell cycle genes including those encoding cyclins (*CCNB1*, *CCNB2,* and *CCNE1*) and cyclin-dependent kinases (*CDK1* and *CDK2*). These results are consistent with prior studies which have reported arrest at the G1/S transition via downregulation of cell cycle regulators [46]. Additionally, we also found increased expression of *GDF-15*, previously identified as one of the most up-regulated genes in response to CBD treatment [47]. It is hypothesised that upregulation of *GDF-15* combined with decreased expression of *FOXM1* act together to form a CBD-dependent antiproliferative pathway across numerous cancer types [47]. The present results provide evidence for the first time that the cannabinoids CBD and CBG have anti-proliferative effects in mesothelioma cells that are associated with cell cycle arrest.

As escape from programmed cell death or apoptosis is an important survival mechanism used by cancers, we also investigated whether CBD and CBG could induce apoptosis in mesothelioma cells. Both CBD and CBG induced apoptosis in all three mesothelioma cell lines which corresponded with expression changes in intrinsic apoptotic pathway genes, including increased expression of *BBC3* (which encodes BCL2 binding component 3) and decreased expression of *BIRC5* (encoding baculoviral inhibitor of apoptosis (IAP) repeat containing 5). Our results agree with numerous prior studies that have reported both CBD and CBG have pro-apoptotic effects in various cancer cell lines [48,49,50]. However, the pro-apoptotic effects of CBD and CBG observed here did not correspond with any increase in reactive oxygen species (ROS) (Appendix A), as has been shown in some prior studies [51]. Autophagy is another cellular process induced in many cancers by phytocannabinoids including CBD [18,52]. Interestingly, in our study autophagy was not induced by either CBD or CBG (Appendix A), contrary to most cancer studies. However, mesothelioma cells have been noted to have high basal levels of autophagy [53] and, therefore, induction may be difficult to identify.

Although distant metastasis in mesothelioma is rare, locally invasive disease occurs in approximately 50% of patients [54,55], contributing to the high mortality of the disease. Therefore, new therapies which can also slow metastatic spread and local invasion are highly sought after. Our results are similar to several other cancer studies, showing that CBD has potent anti-migratory and anti-invasive potential in vitro and in vivo [56,57,58]. Our results here provide only the second report that CBG has anti-invasive activity, which was recently reported using glioblastoma cell lines [59]. To the best of our knowledge this is the first report that CBG has anti-migratory effects in vitro in any type of cancer. The mechanism responsible for these effects is unknown, however we did observe both CBD and CBG treatment reduced expression of the pro-metastatic gene *ID1* (inhibitor of DNA binding 1) in mesothelioma cells. CBD has previously been shown to decrease expression of *ID1* in various cancers [60]. Future studies are needed to probe the mechanism of action of the anti-migratory and anti-invasive effects of the cannabinoids in mesothelioma.

The mechanism of action for phytocannabinoids CBD and CBG is complex with multiple different receptors proposed to be involved [18]. To provide some preliminary insight into the possible mode of action of CBD and CBG, we analysed whether treatment with these cannabinoids altered mRNA expression of various targets implicated in the effects of CBD and CBG. We found that both CBD and CBG produced a dramatic increase in the expression of genes encoding the cannabinoid CB1 receptor, GPR55, and the 5-HT1A receptor in all three mesothelioma cell lines. Moreover, we observed that the cannabinoids also increased the mRNA expression of the chemokine receptor *CXCR4*. Notably, a prior study showed the combination of CBD and THC was required to inhibit expression of *CXCR4* that was associated with anti-migratory effects in multiple myeloma cells [61]. Here, we observed induction of *CXCR4* following treatment with higher cannabinoid concentrations, however different effects on *CXCR4* expression might be observed at lower cannabinoid concentrations that reduce migration and invasion.

We then performed gene pathway analyses to further probe possible mechanisms associated with CBD and CBG treatment on mesothelioma cell lines. Not surprisingly, pathways associated with regulation of the cell cycle were consistently affected by both CBD and CBG. Pathway analysis also revealed disruption of cellular calcium homeostasis through stimulation of Gαq and PLC signalling. These results may be related to the cannabinoid-induced upregulation of *GPR55* we observed here, as GPR55 signals through Gαq and PLC [62]. Moreover, CBG was found to stimulate nuclear factor of activated T cells (NFAT) signalling, a group of transcription factors which is also activated via GPR55 [62]. Both CBD and CBG consistently activated NF-κB, another transcription factor linked to the expression of various inflammatory mediators, including cytokines and chemokines (including CXCL12-CXCR4), but also cell cycle regulators, anti-apoptotic factors, and adhesion molecules [63,64]. Intriguingly, CBD and CBG also affected senescence pathways in both human mesothelioma cell lines, which warrants more detailed examination in future studies.

Given our in vitro data showing that CBD and CBG have potent anti-tumour effects in vitro, we examined the effect of both cannabinoids in a syngeneic orthotopic rat model of mesothelioma [22]. This model is an immunocompetent model that takes into account potential cannabinoid effects on the immune microenvironment, thus offering an advantage over prior research which has largely utilised immunodeficient animals. To provide greater clinical relevance, we also administered doses of the cannabinoids that equate to maximal plasma levels of exposure observed in humans following high CBD doses [31,32]. We were unable to show any survival benefit for rats treated with daily i.p. doses of CBD (40 mg/kg) or CBG (100 mg/kg) in the II45 rat model. This is in contrast to previous studies investigating CBD in mouse models of other cancer types, such as glioma, breast, and lung, which have all demonstrated that CBD reduces tumour growth and metastasis in vivo [65,66]. Although observing reductions in tumour volume is important, the examination of survival in preclinical rodent models of cancer is arguably a more clinically relevant endpoint. Consistent with our present results, a study that examined the effects of CBD (100 mg/kg i.p. administered daily) in a mouse pancreatic cancer xenograft model failed to show any statistically significant improvement in survival [67].

Studies that have demonstrated reduced tumour volumes in mouse xenograft models required repeated dosing of CBD (typically near daily or daily dosing) at between 1 and 25 mg/kg i.p. When considering interspecies differences in body surface area, this amounts to CBD doses of 0.5–12.5 mg/kg in rats, which are lower than the dose used in the present study. CBG has not been widely investigated in vivo, however one study demonstrated that CBG (3 and 10 mg/kg i.p. administered once daily for 5 days) slows tumour growth in a mouse xenograft model of colorectal cancer [51]. Again, these doses are much lower than the dose used in the present study. The argument could then be made that our cannabinoid doses were too high. However, the predicted plasma concentrations that were likely attained here remained 3-fold lower than the IC50 concentrations required to reduce proliferation in vitro. It is therefore likely we were unable to reach a sufficient therapeutic level to produce a significant survival benefit. Our findings highlight the potential difficulty of administering cannabinoids in sufficient quantity for therapeutic benefit in in vivo models of mesothelioma. Future studies may benefit from the use of alternative methods of drug delivery to maximise therapeutic exposures, such as using novel formulation strategies, for example lipid nanoparticles [68]. Overall, the lack of survival benefit seen in our rat model may be due to the aggressive and rapid growth of the II45 cell line in vivo, combined with a failure to reach sufficient drug plasma concentrations.

## 5. Conclusions

Our data present the first report that plant cannabinoids have anti-proliferative effects on mesothelioma cells, that was associated with apoptosis, rather than autophagy or production of ROS. CBD and CBG were the most potent cannabinoids and also inhibited mesothelioma cell migration and invasion. We were unable to show an anti-tumour effect in vivo, potentially due to insufficient plasma concentrations being reached. Thus alternative drug delivery methods may be needed to clinically translate these findings.

## Figures and Tables

**Figure 1 cancers-14-03813-f001:**
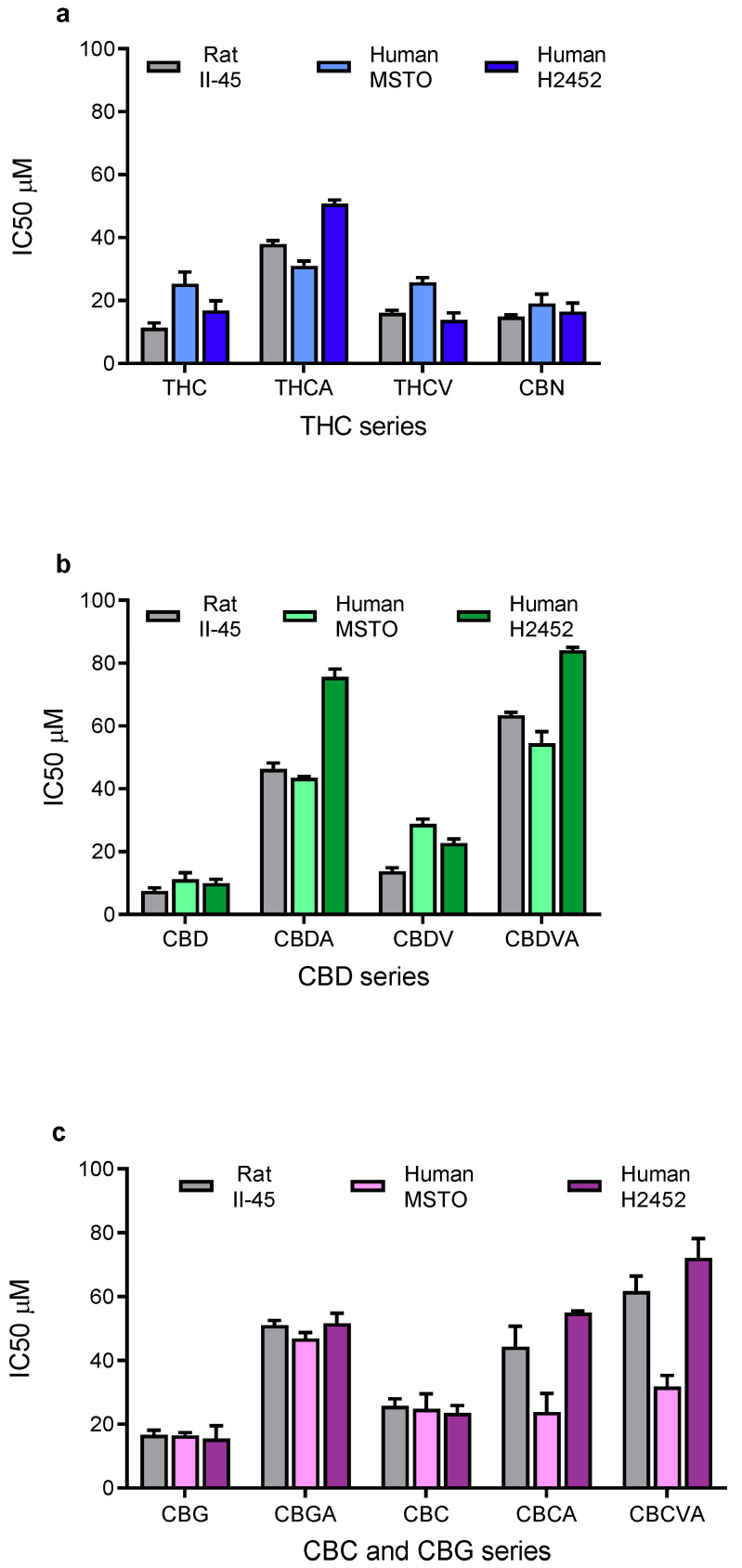
**Phytocannabinoids reduce the proliferation of mesothelioma cell lines.** The drug concentration causing 50% growth inhibition (IC50) of (**a**) ∆^9^-tetrahydrocannabinol (THC) related, (**b**) cannabidiol (CBD) related, and (**c**) cannabichromene (CBC) and cannabigerol (CBG) related phytocannabinoids against rat II-45 and human MSTO and H2452 mesothelioma cell lines were determined. Cell viability was assessed using MTT assays in the presence of phytocannabinoid as indicated. Bars show the mean and standard deviation (SD) from at least three independent experiments.

**Figure 2 cancers-14-03813-f002:**
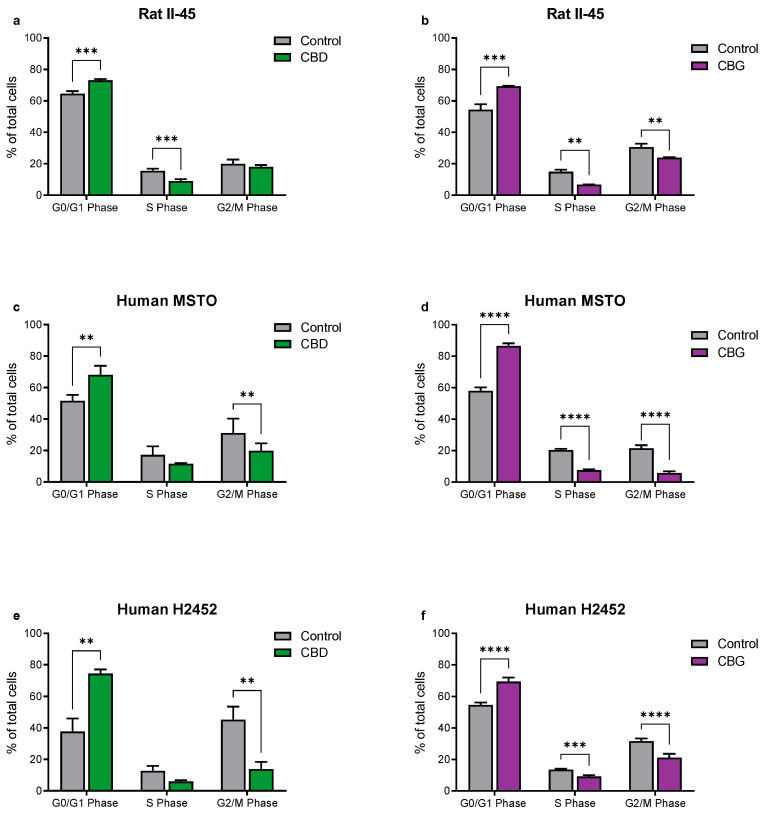
**The phytocannabinoids cannabidiol (CBD) and cannabigerol (CBG) induce cell cycle arrest in mesothelioma cell lines.** The effects of CBD and CBG (2xIC50) on cell cycle were assessed in rat II-45 (**a**,**b**), human MSTO (**c**,**d**) and H2452 (**e**,**f**) mesothelioma cell lines. After 24 h of treatment with vehicle (control), CBD or CBG, cells were harvested, and assays performed. Bars show the mean and SD from three independent experiments. *p*-values were calculated using two-way ANOVA Holm-Sidak’s multiple comparisons. ** *p* < 0.01, *** *p* < 0.001, and **** *p* < 0.0001 relative to vehicle-treated control cells.

**Figure 3 cancers-14-03813-f003:**
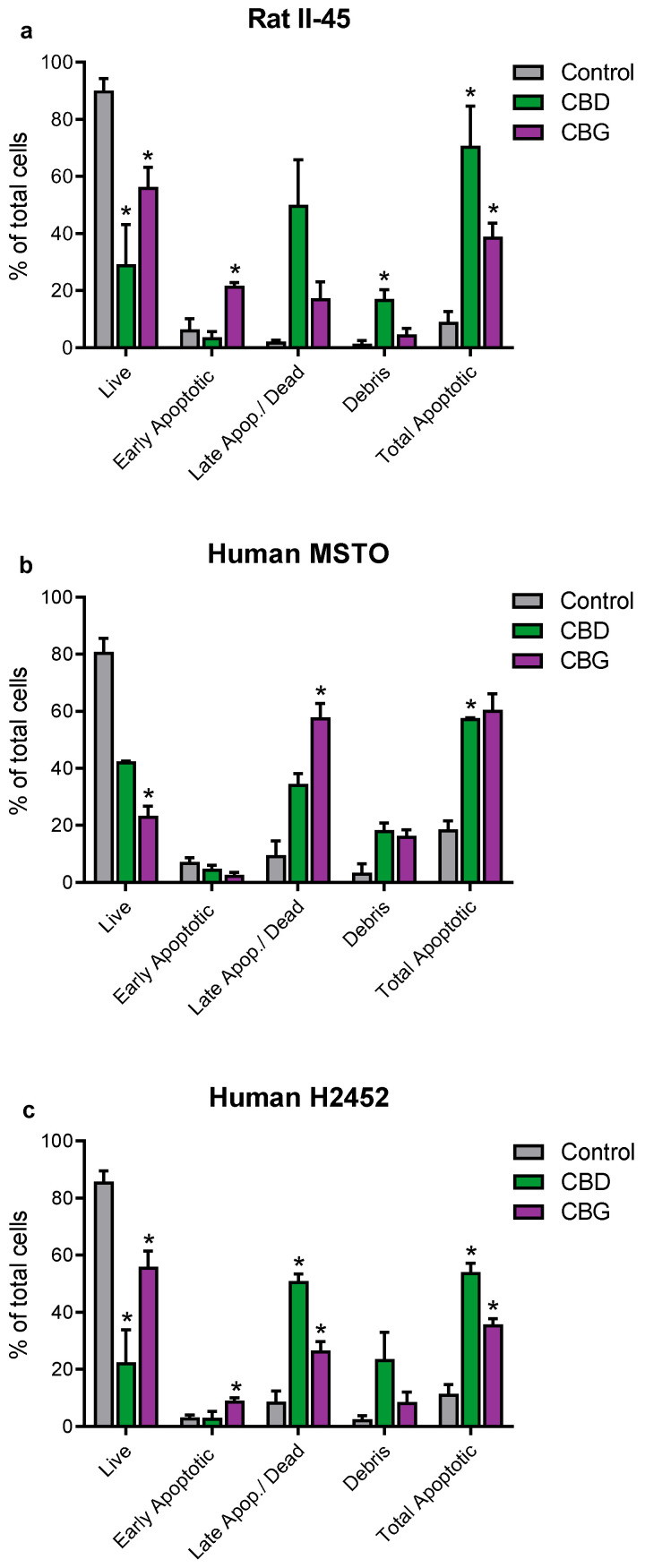
**The phytocannabinoids cannabidiol (CBD) and cannabigerol (CBG) induced apoptosis in mesothelioma cell lines.** The effects of CBD and CBG (2xIC_50_) on apoptosis were assessed in rat II-45 (**a**), human MSTO (**b**), and H2452 (**c**) mesothelioma cell lines. After 24 h of treatment with vehicle (control), CBD or CBG, cells were harvested, and assays performed. Bars show the mean and SD from three independent experiments. *p*-values were calculated using two-way ANOVA Holm-Sidak’s multiple comparisons. * *p* < 0.05 relative to vehicle-treated control cells.

**Figure 4 cancers-14-03813-f004:**
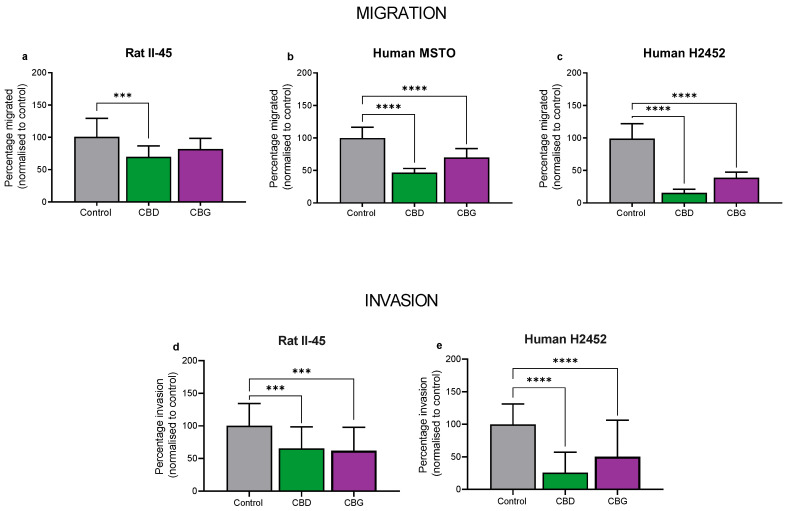
**Cannabidiol (CBD) and cannabigerol (CBG) reduced migration and invasion of mesothelioma cell lines.** Rat II-45 (**a**,**d**) and human MSTO (**b**) and H2452 (**c**,**e**) mesothelioma cell lines were examined with CBD and CBG being tested at sub-cytotoxic concentrations (2 µM CBD for II-45, 6 µM for MSTO and H2452 cells, and 10 µM CBG for all cell lines). The number of migrated cells or cells that had invaded through the Matrigel per 5–10 fields of view was counted and analysed using CellProfiler. Bars show the mean and SD from three independent experiments. *p*-values were calculated using one-way ANOVA Kruskal-Wallis multiple comparisons. *** *p* < 0.001 and **** *p* < 0.0001 relative to vehicle treated control cells.

**Figure 5 cancers-14-03813-f005:**
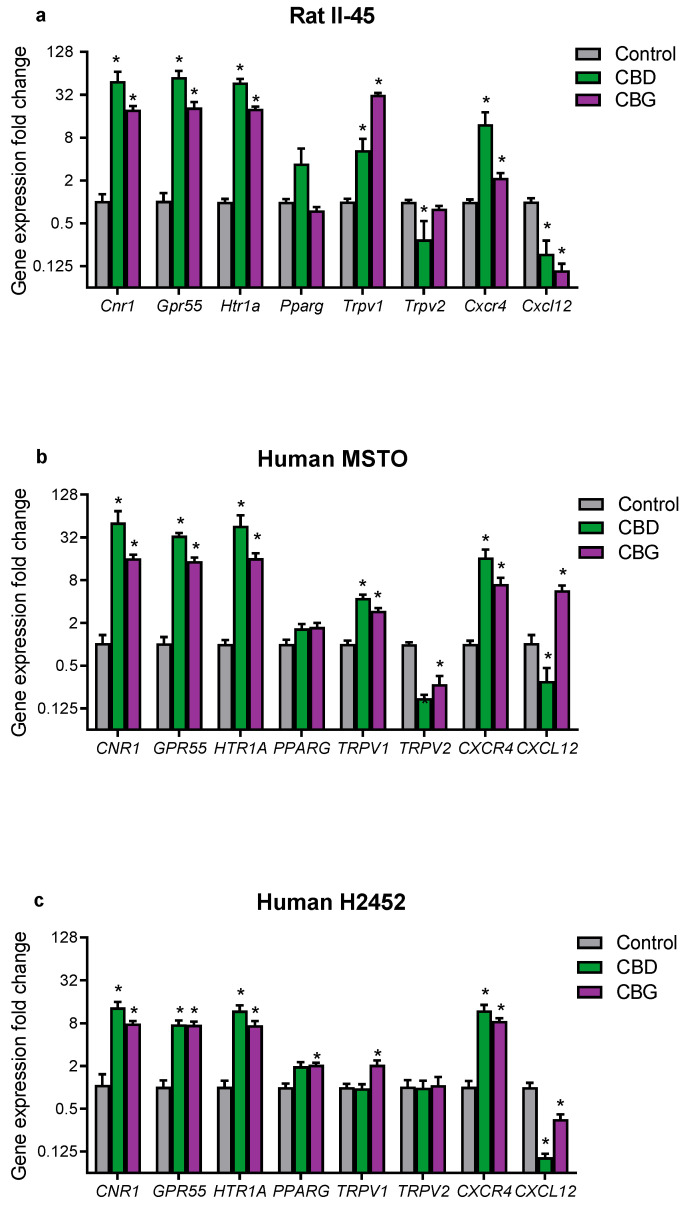
**The phytocannabinoids cannabidiol (CBD) and cannabigerol (CBG) affected mRNA expression of cannabinoid and mesothelioma-related targets in cultured mesothelioma cells.** Gene expression was quantified by RT-qPCR in (**a**) rat II-45 and human (**b**) MSTO and (**c**) H2452 mesothelioma cells. FC in gene expression was calculated relative to untreated control cells using the 2^-delta-delta^ Ct method after normalizing to TBP. Bars show the means and SD in gene expression relative to control (*n* = 3). Grey bars represent a fold change (FC) of one in control treated cells. *p*-values were calculated using two-way ANOVA Holm–Sidak’s multiple comparisons. * *p* < 0.01 relative to vehicle treated control cells and FC < 0.5 or >2.

**Figure 6 cancers-14-03813-f006:**
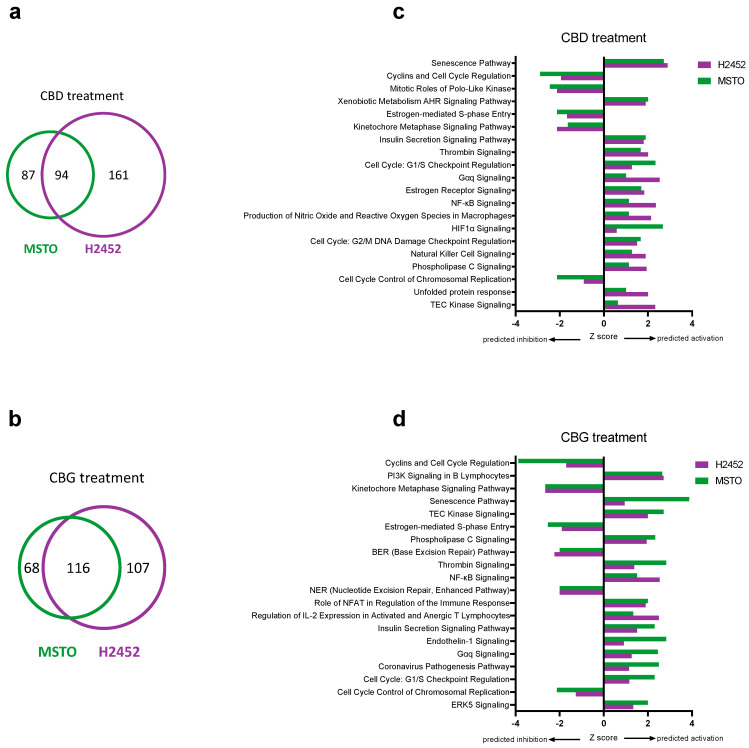
**Differentially expressed genes and pathway analysis associated with cannabidiol (CBD) and cannabigerol (CBG) treatment.** The number of genes found to be differentially expressed (**a**,**b**) and top shared canonical pathways (**c,d**) as a result of CBD (**a**,**c**) and CBG (**b**,**d**) treatment in human MSTO and H2452 mesothelioma cell lines. Analysis was performed relative to vehicle treated control cells and only includes genes with FC > 2. Overlapping areas in Venn diagrams (**a**,**b**) indicate the number of shared genes found to be differentially expressed in the same direction in both cell lines. Z-scores (**c**,**d**) indicate predicted activation state of the top significantly enriched canonical pathways with values < 0 indicating inhibition and values > 0 indicating activation.

**Figure 7 cancers-14-03813-f007:**
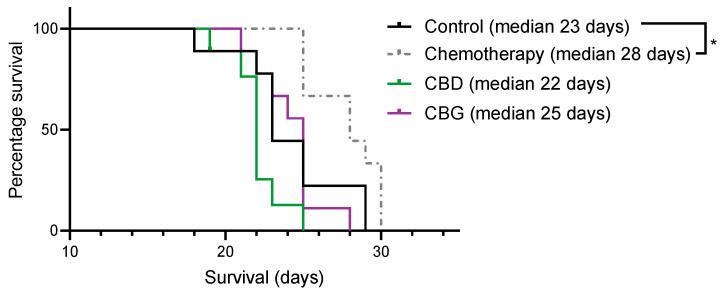
**Pemetrexed + cisplatin but not cannabidiol (CBD) or cannabigerol (CBG) treatment prolongs survival in a rat syngeneic orthotopic model of pleural mesothelioma.** Rats were pleurally engrafted with mesothelioma cells and then treated with control (vehicle), cisplatin + pemetrexed (chemotherapy), 40 mg/kg CBD or 100 mg/kg CBG for 4 weeks and overall survival was assessed. Data were analysed using Log-rank (Mantel-Cox) test. * *p* < 0.05 versus vehicle control.

## Data Availability

The data presented in this study are available in the article and Appendix A. Raw data can be provided upon reasonable request.

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
