# Peer review of "An Examination of the Anti-Cancer Properties of Plant Cannabinoids in Preclinical Models of Mesothelioma"

_cancers, 2022, doi:10.3390/cancers14153813_

Round 1
Reviewer 1 Report
The paper by Colvin et al., described the effect of two phytocannabinoids such as CBD and CBG on mesothelioma. The main finding of the paper is the discrepancy between the results obtained in vitro and in vivo. In particular, the inhibitory effect of CBD and CBG on in vitro mesothelioma cells was convincing and they conformed to previous reports. Although the author made an excellent genetic approach to find new gene regulation end pathway identification, the results mainly highlighted the upregulation of common pathways such as CB1 TRPV1 and GPR55. The interesting data on chemokine receptor CXCR4 is still controversial, as explained by the authors in the discussion section. Unfortunately, the in vivo model did not replicate the effect found on the cell lines. This is probably due to the CBD and CBG’s delivery trouble.
The data presented are interesting and the structure of the paper is acceptable. However, the authors should address some issues regarding the experiments.
- Why did the authors not evaluate the tumor dimension? The overall survival of the animal from chemotherapy and CBG is not so big.
- Did the authors find some other possible pathway involving other genes that resulted up or downregulated such as DTL and GDF-15.
- The results section must be improved
- Spellcheck is necessary throughout the manuscript: i.e. the word cannabidiol is wrong in the title
Author Response
Response to Reviewer 1 Comments
Point 1: Why did the authors not evaluate the tumor dimension? The overall survival of the animal from chemotherapy and CBG is not so big.
Response 1: We were unable to directly measure tumour dimensions throughout the study as tumours were injected orthotopically (into the pleural cavity), rather than subcutaneously. However, severity of pleural dissemination was measured at ethical endpoint, and at this time, all tumours appeared similar. This has now been added to both the methods “Tumours were scored on a scale of 1 to 5 based on volume and infiltration of tumour into the pleural cavity as described previously [34], with 1 indicating the tumor occupied up to 10% of the pleural cavity and 5 indicating the tumor occupied 40–50% of the pleural cavity.” and in the results section “While survival was the primary endpoint of this study, severity of pleural dissemination at endpoint was also assessed. No statistical differences were observed between the groups (data not shown).”
The median overall survival difference between control and chemotherapy treated animals was 5 days. As a percentage of the control, this is an increase of 21.7%, which is a substantial increase in this aggressive and rapid mesothelioma model. We did not directly compare chemotherapy treated and CBG treated animals, but rather compared all treatment groups to control animals. For clarity, the survival analysis method has now been amended and now reads “For survival analysis, p-values were calculated using Log-rank (Mantel-Cox) test, relative to control treated animals.”
Point 2: Did the authors find some other possible pathway involving other genes that resulted up or downregulated such as DTL and GDF-15.
Response 2: Yes, we found numerous possible pathways involving many other genes, including downregulation of DTL and upregulation of GDF-15 in both CBD and CBG treated human mesothelioma cells. We chose to discuss only a few key findings, supported by the literature, however, we have now mentioned both DTL and GDF-15 at the reviewer’s request. The results section now reads “Of note, CBD and CBG consistently decreased expression of key cell cycle genes including those encoding cyclins (CCB1, CCB2 and CCNE1) and cyclin-dependent kinases (CDK1 and CDK2), as well as the anti-apoptotic gene BIRC5, and pro-metastatic gene ID1 and the DTL gene, involved in an array of cancer promoting pathways including proliferation, migration and invasion [35, 36]. Additionally, CBD and CBG consistently increased expression of pro-apoptotic gene BBC3 and of GDF-15, a gene involved in numerous biological functions and a promising prognostic marker in numerous cancers (reviewed in [37]).” The discussion now reads “Additionally, we also found increased expression of GDF-15, previously identified as one of the most up-regulated genes in response to CBD treatment [57]. It is hypothesised that upregulation of GDF-15 combined with decreased expression of FOXM1 act together to form a CBD-dependent antiproliferative pathway across numerous cancer types [57].”
Point 3: The results section must be improved.
Response 3: We have improved the results section as requested.
Point 4: Spellcheck is necessary throughout the manuscript: i.e. the word cannabidiol is wrong in the title
Response 4: We have endeavored to correct any spelling and grammatical errors.
Reviewer 2 Report
An interesting report that is well presented. Some work is needed to provide justification of dosages used for animal work. Additional proofreading is needed as well to fix some grammar mistakes.
Comments:
Abstract
“A panel of thirteen phytocannabinoids inhibited growth of human (MSTO and H2452) 32 and rat (II-45) mesothelioma cells in vitro, with cannabidiol (CBD) and cannabigerol (CBG) the most potent compounds.” – add “were” before “the most”
“These effects were supported by changes in expression of genes associated with cell cycle, proliferation, and cell movement following CBD or CBG treatment.” – add “the” before “expression”
“RNA expression of known cannabinoid receptors CNR1, GPR55, and 5HT1A also increased with CBD or CBG.” – RNA expression mean the process of translation; I assume you meant to say “expression at RNA level”.
“However, treatment with CBD or CBG was unable to increase survival in vivo.” – you should precede this with explaining what animals you used
“…daily treatment for 4 weeks with 40 mg/kg CBD or 100 mg/kg…” – can you justify the reason for these doses? Why was there 2.5-foldmore CBG? Is it just based on IC50? You wrote in the methods section that was partially based on data “of Riedel et. al. [32] to result in similar plasma concentrations (Cmax = 2-4 µM CBD and 2 µM CBG)” – does this mean that CBG is absorbed 2.5 less than CBD? Do you have data on CBG absorption too?
Were there any other parameters evaluated besides survival in in vivo experiment? Did you notice any changes in the rate of metastases?
Author Response
Response to Reviewer 2 Comments
Point 1: Abstract “A panel of thirteen phytocannabinoids inhibited growth of human (MSTO and H2452) 32 and rat (II-45) mesothelioma cells in vitro, with cannabidiol (CBD) and cannabigerol (CBG) the most potent compounds.” – add “were” before “the most”
Response 1: This has been amended and now reads “A panel of thirteen phytocannabinoids inhibited growth of human (MSTO and H2452) and rat (II-45) mesothelioma cells in vitro, and cannabidiol (CBD) and cannabigerol (CBG) were the most potent compounds”.
Point 2: Abstract “These effects were supported by changes in expression of genes associated with cell cycle, proliferation, and cell movement following CBD or CBG treatment.” – add “the” before “expression”
Response 2: This has been amended.
Point 3: Abstract “RNA expression of known cannabinoid receptors CNR1, GPR55, and 5HT1A also increased with CBD or CBG.” – RNA expression mean the process of translation; I assume you meant to say “expression at RNA level”.
Response 3: Changes have been made and this now reads “Gene expression levels of known cannabinoid receptors CNR1, GPR55, and 5HT1A also increased with CBD or CBG treatment.”
Point 4: Abstract “However, treatment with CBD or CBG was unable to increase survival in vivo.” – you should precede this with explaining what animals you used
Response 4: Changes have been made and this now reads “However, treatment with CBD or CBG in a syngeneic rat mesothelioma model was unable to increase survival.”
Point 5: “…daily treatment for 4 weeks with 40 mg/kg CBD or 100 mg/kg…” – can you justify the reason for these doses? Why was there 2.5-foldmore CBG? Is it just based on IC50? You wrote in the methods section that was partially based on data “of Riedel et. al. [32] to result in similar plasma concentrations (Cmax = 2-4 µM CBD and 2 µM CBG)” – does this mean that CBG is absorbed 2.5 less than CBD? Do you have data on CBG absorption too?
Response 5: Yes, Reidel et al (PMID: 21796370) compared the plasma pharmacokinetics of CBD and CBG in rats following i.p. injections and found that total plasma exposures (AUC 0–∞) were approximately 2.5-fold less in CBG compared to CBD dosed animals. This has now been made explicit in the methods section which now reads “Riedel et. al. [32] compared the plasma pharmacokinetics of CBD and CBG in rats following i.p. injections and found that total plasma exposures (AUC 0–∞) were approximately 2.5-fold less in CBG compared to CBD dosed animals. CBD and CBG doses were thus selected based on extrapolated data from our pharmacokinetic experiment and that of Riedel et. al. [32] to result in similar plasma concentrations (Cmax = 2-4 µM CBD and 2 µM CBG), as well as the maximum concentration able to be diluted in solvent.”
Point 6: Were there any other parameters evaluated besides survival in in vivo experiment? Did you notice any changes in the rate of metastases?
Response 6: While survival was the primary endpoint of this study, severity of pleural dissemination at endpoint was also assessed. Severity of pleural dissemination was looked at by assigning a grade from 1-5 (as previously described PMID: 25141917). There were no differences identified between the groups. For clarity, this has now been added to the methods section “Tumours were scored on a scale of 1 to 5 based on volume and infiltration of tumour into the pleural cavity as described previously [34], with 1 indicating the tumour occupied up to 10% of the pleural cavity and 5 indicating the tumour occupied 40–50% of the pleural cavity.” and in the results section “While survival was the primary endpoint of this study, severity of pleural dissemination at endpoint was also assessed. No statistical differences were observed between the groups (data not shown).”
Reviewer 3 Report
Colvin and colleagues used a panel of human and murine mesothelioma cell lines (in vitro studies) and the rat II-45 cells (in vivo study) to show that the non-psychoactive cannabis constitutes, Cannabidoil and Cannabigerol, exert in vitro pro-apoptotic, anti-migratory and anti-invasive effects. These in vitro effects were accompanied with significant changes in gene expression in multiple pathways. In vivo, administration of these compounds (alone) had no effects on rat survival.
The effects of non-psychoactive cannabis constitutes, particularly Cannabidoil, on the progression of mesothelioma is novel. Thus, the topic of this article is of interest to the readers of Cancers.
The in vivo (not the in vitro) study has two major limitations.
First, it relies on changes on rat survival (not anti-tumour and/or anti-metastasis effects) as the sole outcome. Whilst I agree with the authors “the examination of survival is arguably more clinically relevant”, cancer progression in murine models has different dynamic when compared to human. Thus, it wishful thinking to expect that CBD treatment ALONE could prolong survival rate in the model described. To remedy this limitation, I would suggest to, if possible, perform histological analysis in selected tissues to establish if Cannabidoil or Cannabigerol treatment had any effect on tumour cell growth, proliferation and spread in vivo.
The second limitation is the authors should have tested the effects of CBD or CBG in COMBINATION with chemotherapy. This regime might have improved survival rate!!!
Author Response
Response to Reviewer 3 Comments
Point 1: The in vivo (not the in vitro) study has two major limitations. First, it relies on changes on rat survival (not anti-tumour and/or anti-metastasis effects) as the sole outcome. Whilst I agree with the authors “the examination of survival is arguably more clinically relevant”, cancer progression in murine models has different dynamic when compared to human. Thus, it wishful thinking to expect that CBD treatment ALONE could prolong survival rate in the model described. To remedy this limitation, I would suggest to, if possible, perform histological analysis in selected tissues to establish if Cannabidoil or Cannabigerol treatment had any effect on tumour cell growth, proliferation and spread in vivo.
Response 1: We did not assess histological differences between treatment groups in this study. We have previously looked at this in similar studies and have not noted any differences. It is believed that because the tumours are taken at ethical endpoint (i.e. a time when the tumour cannot progress any further as it is causing life threatening consequences to the animal), no differences would be detectable. A cross sectional analysis at a set timepoint may be a better way to assess changes in tumour histology, however this is beyond the scope of the study.
Point 2: The second limitation is the authors should have tested the effects of CBD or CBG in COMBINATION with chemotherapy. This regime might have improved survival rate!!!
Response 2: The main goal of this paper was to investigate the anti-cancer properties of phytocannabinoids in mesothelioma as single therapies. In the in vivo study, chemotherapy was used as a positive control for increased survival. We agree that it is important to examine CBD and CBG combined with chemotherapy, but we believe that this is a separate study beyond the scope of this paper. Preliminary in vitro data from our lab demonstrated antagonism when chemotherapy was used in combination with CBD or CBG. A very recent study in melanoma, also reports antagonism when using CBD together with cisplatin (PMID: 35743195). We are currently investigating this further in our mesothelioma model.